# FFJORD: FREE-FORM CONTINUOUS DYNAMICS FOR SCALABLE REVERSIBLE GENERATIVE MODELS

**Will Grathwohl**[*†‡] **, Ricky T. Q. Chen**[*†]**, Jesse Bettencourt**[†]**, Ilya Sutskever**[‡] **, David Duvenaud**[†]

## ABSTRACT

Reversible generative models map points from a simple distribution to a complex distribution through an easily invertible neural network. Likelihood-based training of these models requires restricting their architectures to allow cheap computation of Jacobian determinants. Alternatively, the Jacobian trace can be used if the transformation is specified by an ordinary differential equation. In this paper, we use Hutchinson's trace estimator to give a scalable unbiased estimate of the log-density. The result is a continuous-time invertible generative model with unbiased density estimation and one-pass sampling, while allowing unrestricted neural network architectures. We demonstrate our approach on high-dimensional density estimation, image generation, and variational inference, improving the state-of-the-art among exact likelihood methods with efficient sampling.

## 1 INTRODUCTION

Reversible generative models use cheaply invertible neural networks to transform samples from a fixed base distribution. Examples include NICE (Dinh et al., 2014), Real NVP (Dinh et al., 2017), and Glow (Kingma & Dhariwal, 2018). These models are easy to sample from, and can be trained by maximum likelihood using the change of variables formula. However, this requires placing awkward restrictions on their architectures, such as partitioning dimensions or using rank one weight matrices, in order to avoid an $\mathcal{O}(D^3)$ cost determinant computation.

Recently, Chen et al. (2018) introduced continuous normalizing flows (CNF), defining the mapping from latent variables to data using ordinary differential equations (ODE). In their model, the likelihood can be computed using trace operations costing only $\mathcal{O}(D^2)$. This allows a more flexible, but still restricted, family of network architectures to be used.

Extending this work, we introduce an unbiased stochastic estimator of the likelihood that has $\mathcal{O}(D)$ time cost, allowing completely unrestricted architectures. Furthermore, we have implemented GPU-based adaptive ODE solvers to train and evaluate these models on modern hardware. We call our approach Free-form Jacobian of Reversible Dynamics (FFJORD). Figure 1 shows FFJORD smoothly transforming a Gaussian distribution into a multi-modal distribution.

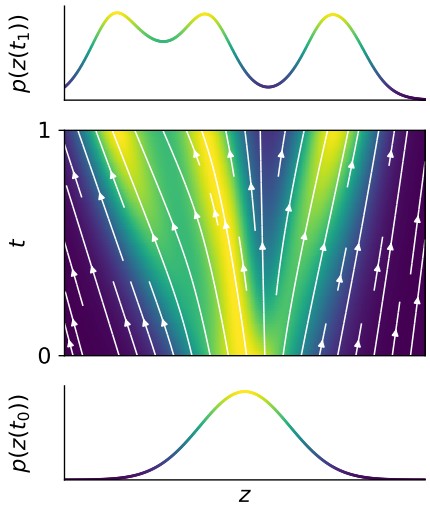

Figure 1: FFJORD transforms a simple base distribution at $t_0$ into the target distribution at $t_1$ by integrating over learned continuous dynamics.

---

[*]Equal contribution. Order determined by coin toss. {wgrathwohl, rtqichen}@cs.toronto.edu
[†]University of Toronto and Vector Institute. [‡]OpenAI.

## 2 BACKGROUND: GENERATIVE MODELS AND CHANGE OF VARIABLES

In contrast to directly parameterizing a normalized distribution (e.g. Oord et al. (2016); Germain et al. (2015)), the change of variables formula allows one to specify a complex normalized distribution $p_\mathbf{x}(\mathbf{x})$ implicitly by warping a normalized base distribution $p_\mathbf{z}(\mathbf{z})$ through an invertible function $f : \mathbb{R}^D \to \mathbb{R}^D$. Given a random variable $\mathbf{z} \sim p_\mathbf{z}(\mathbf{z})$ the log density of $\mathbf{x} = f(\mathbf{z})$ follows

$$\log p_\mathbf{x}(\mathbf{x}) = \log p_\mathbf{z}(\mathbf{z}) - \log \det \left| \frac{\partial f(\mathbf{z})}{\partial \mathbf{z}} \right| \tag{1}$$

where $\partial f(\mathbf{z})/\partial \mathbf{z}$ is the Jacobian of $f$. In general, computing the log determinant has a time cost of $\mathcal{O}(D^3)$. Much work has gone into developing restricted neural network architectures which make computing the Jacobian's determinant more tractable. These approaches broadly fall into three categories:

**Normalizing flows.** By restricting the functional form of $f$, various determinant identities can be exploited (Rezende & Mohamed, 2015; Berg et al., 2018). These models cannot be trained as generative models from data because they do not have a tractable inverse $f^{-1}$. However, they are useful for specifying approximate posteriors for variational inference (Kingma & Welling, 2014).

**Autoregressive transformations.** By using an autoregressive model and specifying an ordering of the dimensions, the Jacobian of $f$ is enforced to be lower triangular (Kingma et al., 2016; Oliva et al., 2018). These models excel at density estimation for tabular datasets (Papamakarios et al., 2017), but require $D$ sequential evaluations of $f$ to invert, which is prohibitive when $D$ is large.

**Partitioned transformations.** Partitioning the dimensions and using affine transformations makes the determinant of the Jacobian cheap to compute, and the inverse $f^{-1}$ computable with the same cost as $f$ (Dinh et al., 2014; 2017). This method allows the use of convolutional architectures, excelling at density estimation for image data (Dinh et al., 2017; Kingma & Dhariwal, 2018).

Throughout this work, we refer to *reversible generative models* as those which use the change of variables to transform a base distribution to the model distribution while maintaining both efficient density estimation and efficient sampling capabilities using a single pass of the model.

### 2.1 OTHER GENERATIVE MODELS

There exist several approaches to generative modeling approaches which do not use the change of variables equation for training. Generative adversarial networks (GANs) (Goodfellow et al., 2014) use large, unrestricted neural networks to transform samples from a fixed base distribution. Lacking a closed-form likelihood, an auxiliary discriminator model must be trained to estimate divergences or density ratios in order to provide a training signal. Autoregressive models (Germain et al., 2015; Oord et al., 2016) directly specify the joint distribution $p(\mathbf{x})$ as a sequence of explicit conditional distributions using the product rule. These models require at least $\mathcal{O}(D)$ evaluations to sample from. Variational autoencoders (VAEs) (Kingma & Welling, 2014) use an unrestricted architecture to explicitly specify the conditional likelihood $p(x|z)$, but can only efficiently provide a stochastic lower bound on the marginal likelihood $p(x)$.

### 2.2 CONTINUOUS NORMALIZING FLOWS

Chen et al. (2018) define a generative model for data $\mathbf{x} \in \mathbb{R}^D$ similar to those based on (1), but replace the warping function with an integral of continuous-time dynamics. The generative process first samples from a base distribution $\mathbf{z}_0 \sim p_{z_0}(\mathbf{z}_0)$. Then, given an ODE whose dynamics are defined by the parametric function $\partial \mathbf{z}(t)/\partial t = f(\mathbf{z}(t), t; \theta)$, we solve the initial value problem with $\mathbf{z}(t_0) = \mathbf{z}_0$ to obtain a data sample $\mathbf{x} = \mathbf{z}(t_1)$. These models are called Continous Normalizing Flows (CNF). The change in log-density under this model follows a second differential equation, called the *instantaneous change of variables* formula (Chen et al., 2018):

$$\frac{\partial \log p(\mathbf{z}(t))}{\partial t} = -\operatorname{Tr}\left( \frac{\partial f}{\partial \mathbf{z}(t)} \right). \tag{2}$$

We can compute total change in log-density by integrating across time:

$$\log p(\mathbf{z}(t_1)) = \log p(\mathbf{z}(t_0)) - \int_{t_0}^{t_1} \operatorname{Tr}\left( \frac{\partial f}{\partial \mathbf{z}(t)} \right) dt. \tag{3}$$

| Method | Train on data | One-pass Sampling | Exact/Unbiased Log-likelihood | Free-form Jacobian |
|---|:---:|:---:|:---:|:---:|
| Variational Autoencoders | ✓ | ✓ | ✗ | ✓ |
| Generative Adversarial Nets | ✓ | ✓ | ✗ | ✓ |
| Likelihood-based Autoregressive | ✓ | ✗ | ✓ | ✗ |
| Normalizing Flows | ✗ | ✓ | ✓ | ✗ |
| Reverse-NF, MAF, TAN | ✓ | ✗ | ✓ | ✗ |
| NICE, Real NVP, Glow, Planar CNF | ✓ | ✓ | ✓ | ✗ |
| **FFJORD** | ✓ | ✓ | ✓ | ✓ |

*(The bottom four rows are grouped under the left-margin label "Change of Variables".)*

Table 1: A comparison of recent generative modeling approaches.

Given a datapoint $\mathbf{x}$, we can compute both the point $\mathbf{z}_0$ which generates $\mathbf{x}$, as well as $\log p(\mathbf{x})$ under the model by solving the combined initial value problem:

$$\underbrace{\begin{bmatrix} \mathbf{z}_0 \\ \log p(\mathbf{x}) - \log p_{z_0}(\mathbf{z}_0) \end{bmatrix}}_{\text{solutions}} = \begin{bmatrix} \mathbf{x} \\ 0 \end{bmatrix} + \underbrace{\int_{t_1}^{t_0} \begin{bmatrix} f(\mathbf{z}(t), t; \theta) \\ \mathrm{Tr}\left( \frac{\partial f}{\partial \mathbf{z}(t)} \right) \end{bmatrix} dt}_{\text{dynamics}}, \qquad \underbrace{\begin{bmatrix} \mathbf{z}(t_1) \\ \log p(\mathbf{x}) - \log p(\mathbf{z}(t_1)) \end{bmatrix}}_{\text{initial values}} = \begin{bmatrix} \mathbf{x} \\ 0 \end{bmatrix}$$

(4)

which integrates the combined dynamics of $z(t)$ and the log-density of the sample backwards in time from $t_1$ to $t_0$. We can then compute $\log p(\mathbf{x})$ using the solution of (4) and adding $\log p_{z_0}(\mathbf{z}_0)$. The existence and uniqueness of (4) require that $f$ and its first derivatives be Lipschitz continuous (Khalil, 2002), which can be satisfied in practice using neural networks with smooth Lipschitz activations, such as softplus or tanh.

### 2.2.1 BACKPROPAGATING THROUGH ODE SOLUTIONS WITH THE ADJOINT METHOD

CNFs are trained to maximize (3). This objective involves the solution to an initial value problem with dynamics parameterized by $\theta$. For any scalar loss function which operates on the solution to an initial value problem

$$L(\mathbf{z}(t_1)) = L\left( \int_{t_0}^{t_1} f(\mathbf{z}(t), t; \theta) dt \right)$$

(5)

then Pontryagin (1962) shows that its derivative takes the form of another initial value problem

$$\frac{dL}{d\theta} = -\int_{t_1}^{t_0} \left( \frac{\partial L}{\partial \mathbf{z}(t)} \right)^T \frac{\partial f(\mathbf{z}(t), t; \theta)}{\partial \theta} dt.$$

(6)

The quantity $-\partial L / \partial \mathbf{z}(t)$ is known as the adjoint state of the ODE. Chen et al. (2018) use a black-box ODE solver to compute $\mathbf{z}(t_1)$, and then a separate call to a solver to compute (6) with the initial value $\partial L / \partial \mathbf{z}(t_1)$. This approach is a continuous-time analog to the backpropgation algorithm (Rumelhart et al., 1986; Andersson, 2013) and can be combined with gradient-based optimization to fit the parameters $\theta$ by maximum likelihood.

## 3 SCALABLE DENSITY EVALUATION WITH UNRESTRICTED ARCHITECTURES

Switching from discrete-time dynamics to continuous-time dynamics reduces the primary computational bottleneck of normalizing flows from $\mathcal{O}(D^3)$ to $\mathcal{O}(D^2)$, at the cost of introducing a numerical ODE solver. This allows the use of more expressive architectures. For example, each layer of the original normalizing flows model of Rezende & Mohamed (2015) is a one-layer neural network with only a single hidden unit. In contrast, the instantaneous transformation used in planar continuous normalizing flows (Chen et al., 2018) is a one-layer neural network with many hidden units. In this section, we construct an unbiased estimate of the log-density with $\mathcal{O}(D)$ cost, allowing completely unrestricted neural network architectures to be used.

## 3.1 Unbiased Linear-time Log-Density Estimation

In general, computing $\text{Tr}\left(\partial f / \partial \mathbf{z}(t)\right)$ exactly costs $\mathcal{O}(D^2)$, or approximately the same cost as $D$ evaluations of $f$, since each entry of the diagonal of the Jacobian requires computing a separate derivative of $f$ (Griewank & Walther, 2008). However, there are two tricks that can help. First, vector-Jacobian products $\boldsymbol{v}^T \frac{\partial f}{\partial \mathbf{z}}$ can be computed for approximately the same cost as evaluating $f$ using reverse-mode automatic differentiation. Second, we can get an unbiased estimate of the trace of a matrix by taking a double product of that matrix with a noise vector:

$$\text{Tr}(A) = \mathbb{E}_{p(\boldsymbol{\epsilon})}[\boldsymbol{\epsilon}^T A \boldsymbol{\epsilon}]. \tag{7}$$

The above equation holds for any $D$-by-$D$ matrix $A$ and distribution $p(\boldsymbol{\epsilon})$ over $D$-dimensional vectors such that $\mathbb{E}[\boldsymbol{\epsilon}] = 0$ and $\text{Cov}(\boldsymbol{\epsilon}) = I$. The Monte Carlo estimator derived from (7) is known as Hutchinson's trace estimator (Hutchinson, 1989; Adams et al., 2018).

To keep the dynamics deterministic within each call to the ODE solver, we can use a fixed noise vector $\boldsymbol{\epsilon}$ for the duration of each solve without introducing bias:

$$
\begin{aligned}
\log p(\mathbf{z}(t_1)) &= \log p(\mathbf{z}(t_0)) - \int_{t_0}^{t_1} \text{Tr}\left(\frac{\partial f}{\partial \mathbf{z}(t)}\right) dt \\
&= \log p(\mathbf{z}(t_0)) - \int_{t_0}^{t_1} \mathbb{E}_{p(\boldsymbol{\epsilon})}\left[\boldsymbol{\epsilon}^T \frac{\partial f}{\partial \mathbf{z}(t)} \boldsymbol{\epsilon}\right] dt \\
&= \log p(\mathbf{z}(t_0)) - \mathbb{E}_{p(\boldsymbol{\epsilon})}\left[\int_{t_0}^{t_1} \boldsymbol{\epsilon}^T \frac{\partial f}{\partial \mathbf{z}(t)} \boldsymbol{\epsilon} dt\right]
\end{aligned}
\tag{8}
$$

Typical choices of $p(\boldsymbol{\epsilon})$ are a standard Gaussian or Rademacher distribution (Hutchinson, 1989).

### 3.1.1 Reducing Variance with Bottleneck Capacity

Often, there exist bottlenecks in the architecture of the dynamics network, i.e. hidden layers whose width $H$ is smaller than the dimensions of the input $D$. In such cases, we can reduce the variance of Hutchinson's estimator by using the cyclic property of trace. Since the variance of the estimator for $\text{Tr}(A)$ grows asymptotic to $||A||_F^2$ (Hutchinson, 1989), we suspect that having fewer dimensions should help reduce variance. If we view the dynamics as a composition of two functions $f = g \circ h(\mathbf{z})$ then we observe

$$\text{Tr}\underbrace{\left(\frac{\partial f}{\partial \mathbf{z}}\right)}_{D \times D} = \text{Tr}\underbrace{\left(\frac{\partial g}{\partial h} \frac{\partial h}{\partial \mathbf{z}}\right)}_{D \times D} = \text{Tr}\underbrace{\left(\frac{\partial h}{\partial \mathbf{z}} \frac{\partial g}{\partial h}\right)}_{H \times H} = \mathbb{E}_{p(\boldsymbol{\epsilon})}\left[\boldsymbol{\epsilon}^T \frac{\partial h}{\partial \mathbf{z}} \frac{\partial g}{\partial h} \boldsymbol{\epsilon}\right]. \tag{9}$$

When $f$ has multiple hidden layers, we choose $H$ to be the smallest dimension. This bottleneck trick can reduce the norm of the matrix which may also help reduce the variance of the trace estimator. As introducing a bottleneck limits our model capacity, we do not use this trick in our experiments. However this trick can reduce variance when a bottleneck is used, as shown in our ablation studies.

## 3.2 FFJORD: A Continuous-time Reversible Generative Model

Our complete method uses the dynamics defined in (2) and the efficient log-likelihood estimator of (8) to produce the first scalable and reversible generative model with an unconstrained Jacobian. We call this method Free-Form Jacobian of Reversible Dyanamics (FFJORD). Pseudo-code of our method is given in Algorithm 1, and Table 1 summarizes the capabilities of our model compared to other recent generative modeling approaches.

Assuming the cost of evaluating $f$ is on the order of $\mathcal{O}(DH)$ where $D$ is the dimensionality of the data and $H$ is the size of the largest hidden layer in $f$, then the cost of computing the likelihood in models with repeated use of invertible transformations (1) is $\mathcal{O}((DH + D^3)L)$ where $L$ is the number of transformations used. For CNF, this reduces to $\mathcal{O}((DH + D^2)\hat{L})$ for CNFs, where $\hat{L}$ is the number of evaluations of $f$ used by the ODE solver. With FFJORD, this reduces further to $\mathcal{O}((DH + D)\hat{L})$.

---

**Algorithm 1** Unbiased stochastic log-density estimation using the FFJORD model

---

**Require:** dynamics $f_\theta$, start time $t_0$, stop time $t_1$, data samples $\mathbf{x}$, data dimension $D$.
    $\epsilon \leftarrow$ sample_unit_variance($\mathbf{x}$.shape)            $\triangleright$ Sample $\epsilon$ outside of the integral
    **function** $f_{aug}([\mathbf{z}_t, \log p_t], t)$:            $\triangleright$ Augment $f$ with log-density dynamics.
        $f_t \leftarrow f_\theta(\mathbf{z}(t), t)$            $\triangleright$ Evaluate neural network
        $g \leftarrow \epsilon^T \frac{\partial f}{\partial \mathbf{z}}\big|_{\mathbf{z}(t)}$       $\triangleright$ Compute vector-Jacobian product with automatic differentiation
        $\widetilde{\text{Tr}} = g\epsilon$         $\triangleright$ Unbiased estimate of $\text{Tr}(\frac{\partial f}{\partial \mathbf{z}})$ with $\epsilon^T \frac{\partial f}{\partial \mathbf{z}} \epsilon$
        **return** $[f_t, -\widetilde{\text{Tr}}]$       $\triangleright$ Concatenate dynamics of state and log-density
    **end function**
    $[\mathbf{z}_0, \Delta_{\log p}] \leftarrow$ odeint($f_{\text{aug}}, [\mathbf{x}, \vec{0}], t_0, t_1$)   $\triangleright$ Solve the ODE $\int_{t_0}^{t_1} f_{\text{aug}}([\mathbf{z}(t), \log p(\mathbf{z}(t))], t) \, dt$
    $\log \hat{p}(\mathbf{x}) \leftarrow \log p_{\mathbf{z}_0}(\mathbf{z}_0) - \Delta_{\log p}$         $\triangleright$ Add change in log-density
    **return** $\log \hat{p}(\mathbf{x})$

---

## 4 EXPERIMENTS

We demonstrate FFJORD on a variety of density estimation tasks, and for approximate inference in variational autoencoders (Kingma & Welling, 2014). Experiments were conducted using a suite of GPU-based ODE-solvers and an implementation of the adjoint method for backpropagation[1]. In all experiments the Runge-Kutta 4(5) algorithm with the tableau from Shampine (1986) was used to solve the ODEs. We ensure tolerance is set low enough so numerical error is negligible; see Appendix C.

We used Hutchinson's trace estimator (7) during training and the exact trace when reporting test results. This was done in all experiments except for our density estimation models trained on MNIST and CIFAR10 where computing the exact Jacobian trace was too expensive.

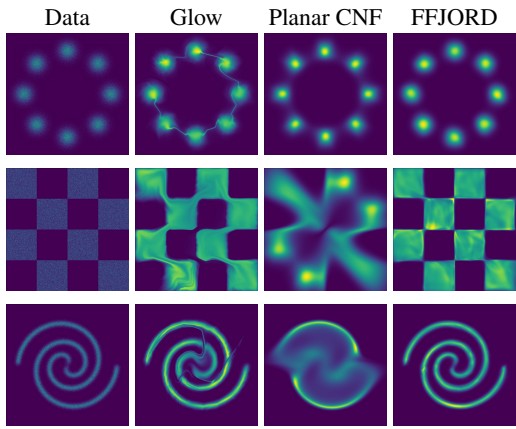

Figure 2: Comparison of trained Glow, planar CNF, and FFJORD models on 2-dimensional distributions, including multi-modal and discontinuous densities.

The dynamics of FFJORD are defined by a neural network $f$ which takes as input the current state $\mathbf{z}(t) \in \mathbb{R}^D$ and the current time $t \in \mathbb{R}$. We experimented with several ways to incorporate $t$ as an input to $f$, such as hyper-networks, but found that simply concatenating $t$ on to $\mathbf{z}(t)$ at the input to every layer worked well and was used in all of our experiments.

### 4.1 DENSITY ESTIMATION ON TOY 2D DATA

We first train on 2 dimensional data to visualize the model and the learned dynamics.[2] In Figure 2, we show that by warping a simple isotropic Gaussian, FFJORD can fit both *multi-modal* and even *discontinuous* distributions. The number of evaluations of the ODE solver is roughly 70-100 on all datasets, so we compare against a Glow model with 100 discrete layers.

The learned distributions of both FFJORD and Glow can be seen in Figure 2. Interestingly, we find that Glow learns to stretch the unimodal base distribution into multiple modes but has trouble modeling the areas of low probability between disconnected regions. In contrast, FFJORD is capable of modeling disconnected modes and can also learn convincing approximations of discontinuous density functions (middle row in Figure 2). Since the main benefit of FFJORD is the ability to train with deeper dynamics networks, we also compare against planar CNF (Chen et al., 2018) which can

---

[1]Code can be found at https://github.com/rtqichen/ffjord and https://github.com/rtqichen/torchdiffeq.
[2]Videos of the learned dynamics can be found at https://imgur.com/a/Rtr3Mbq.

Samples 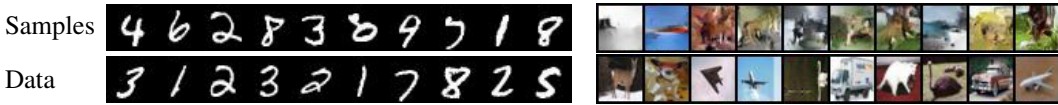
Data

Figure 3: Samples and data from our image models. MNIST on left, CIFAR10 on right.

|  | POWER | GAS | HEPMASS | MINIBOONE | BSDS300 | MNIST | CIFAR10 |
|---|---|---|---|---|---|---|---|
| Real NVP | -0.17 | -8.33 | 18.71 | 13.55 | -153.28 | 1.06* | 3.49* |
| Glow | -0.17 | -8.15 | 18.92 | 11.35 | -155.07 | 1.05* | **3.35*** |
| FFJORD | **-0.46** | **-8.59** | **14.92** | **10.43** | **-157.40** | **0.99*** $(1.05^{\dagger})$ | 3.40* |
| MADE | 3.08 | -3.56 | 20.98 | 15.59 | -148.85 | 2.04 | 5.67 |
| MAF | -0.24 | -10.08 | 17.70 | 11.75 | -155.69 | 1.89 | 4.31 |
| TAN | -0.48 | -11.19 | 15.12 | 11.01 | -157.03 | - | - |
| MAF-DDSF | -0.62 | -11.96 | 15.09 | 8.86 | -157.73 | - | - |

Table 2: Negative log-likelihood on test data for density estimation models; **lower is better**. In nats for tabular data and bits/dim for MNIST and CIFAR10. *Results use multi-scale convolutional architectures. $^{\dagger}$Results use a single flow with a convolutional encoder-decoder architecture.

be viewed as a single hidden layer network. Without the benefit of a flexible network, planar CNF is unable to model complex distributions.

## 4.2 DENSITY ESTIMATION ON REAL DATA

We perform density estimation on five tabular datasets preprocessed as in Papamakarios et al. (2017) and two image datasets; MNIST and CIFAR10. When reproducing Glow, we use the same configurations for Real NVP as Papamakarios et al. (2017) and add invertible fully connected layer between all coupling layers. On the tabular datasets, FFJORD performs the best out of reversible models by a wide margin but is outperformed by recent autoregressive models. Of those, FFJORD outperforms MAF (Papamakarios et al., 2017) on all but one dataset and manages to outperform TAN Oliva et al. (2018) on the MINIBOONE dataset. These models require $\mathcal{O}(D)$ sequential computations to sample from while the best performing method, MAF-DDSF (Huang et al., 2018), cannot be sampled from without resorting to correlated or expensive sampling algorithms such as MCMC.

On MNIST we find that FFJORD can model the data as effectively as Glow and Real NVP using only a single flow defined by a single neural network. This is in contrast to Glow and Real NVP which must compose many flows to achieve similar performance. When we use multiple flows in a multiscale architecture (like those used by Glow and Real NVP) we obtain better performance on MNIST and comparable performance to Glow on CIFAR10. Notably, FFJORD is able to achieve this performance while *using less than 2% as many parameters* as Glow. We also note that Glow uses a learned base distribution whereas FFJORD and Real NVP use a fixed Gaussian. A summary of our results on density estimation can be found in Table 2 and samples can be seen in Figure 3. Full details on architectures used, our experimental procedure, and additional samples can be found in Appendix B.1.

In general, our approach is slower than competing methods, but we find the memory-efficiency of the adjoint method allows us to use much larger batch sizes than those methods. On the tabular datasets we used a batch sizes up to 10,000 and on the image datasets we used a batch size of 900.

## 4.3 VARIATIONAL AUTOENCODER

We compare FFJORD to other normalizing flows for use in variational inference. We train a VAE (Kingma & Welling, 2014) on four datasets using a FFJORD flow and compare to VAEs with no flow, Planar Flows (Rezende & Mohamed, 2015), Inverse Autoregressive Flow (IAF) (Kingma

|  | MNIST | Omniglot | Frey Faces | Caltech Silhouettes |
|---|---|---|---|---|
| No Flow | $86.55 \pm .06$ | $104.28 \pm .39$ | $4.53 \pm .02$ | $110.80 \pm .46$ |
| Planar | $86.06 \pm .31$ | $102.65 \pm .42$ | $4.40 \pm .06$ | $109.66 \pm .42$ |
| IAF | $84.20 \pm .17$ | $102.41 \pm .04$ | $4.47 \pm .05$ | $111.58 \pm .38$ |
| Sylvester | $83.32 \pm .06$ | $99.00 \pm .04$ | $4.45 \pm .04$ | $104.62 \pm .29$ |
| FFJORD | $\mathbf{82.82 \pm .01}$ | $\mathbf{98.33 \pm .09}$ | $\mathbf{4.39 \pm .01}$ | $\mathbf{104.03 \pm .43}$ |

Table 3: Negative ELBO on test data for VAE models; **lower is better**. In nats for all datasets except Frey Faces which is presented in bits per dimension. Mean/stdev are estimated over 3 runs.

et al., 2016), and Sylvester normalizing flows (Berg et al., 2018). To provide a fair comparison, our encoder/decoder architectures and learning setup exactly mirror those of Berg et al. (2018).

In VAEs it is common for the encoder network to also output the parameters of the flow as a function of the input $\mathbf{x}$. With FFJORD, we found this led to differential equations which were too difficult to integrate numerically. Instead, the encoder network outputs a low-rank update to a global weight matrix and an input-dependent bias vector. When used in recognition nets, neural network layers defining the dynamics inside FFJORD take the form

$$\text{layer}(h; \mathbf{x}, W, b) = \sigma\left(\left(\underbrace{W}_{D_{out} \times D_{in}} + \underbrace{\hat{U}(\mathbf{x})}_{D_{out} \times k} \underbrace{\hat{V}(\mathbf{x})^T}_{D_{in} \times k}\right) h + \underbrace{b}_{D_{out} \times 1} + \underbrace{\hat{b}(\mathbf{x})}_{D_{out} \times 1}\right) \qquad (10)$$

where $h$ is the input to the layer, $\sigma$ is an element-wise activation function, $D_{in}$ and $D_{out}$ are the input and output dimension of this layer, and $\hat{U}(\mathbf{x})$, $\hat{V}(\mathbf{x})$, $\hat{b}(\mathbf{x})$ are input-dependent parameters returned from an encoder network. A full description of the model architectures used and our experimental setup can be found in Appendix B.2.

On every dataset tested, FFJORD outperforms all other competing normalizing flows. A summary of our variational inference results can be found in Table 3.

## 5 ANALYSIS AND DISCUSSION

We performed a series of ablation experiments to gain a better understanding of the proposed model.

### 5.1 FASTER TRAINING WITH BOTTLENECK TRICK

We plotted the training losses on MNIST using an encoder-decoder architecture (see Appendix B.1 for details). Loss during training is plotted in Figure 4, where we use the trace estimator directly on the $D \times D$ Jacobian, or we use the bottleneck trick to reduce the dimension to $H \times H$. Interestingly, we find that while the bottleneck trick (9) can lead to faster convergence when the trace is estimated using a Gaussian-distributed $\epsilon$, we did not observe faster convergence when using a Rademacher-distributed $\epsilon$.

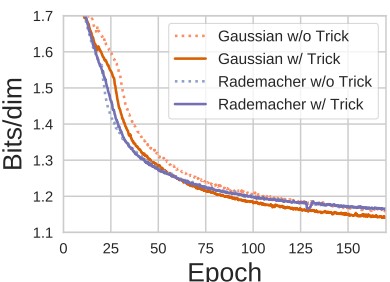

Figure 4: The variance of our model's log-density estimator can be reduced using neural network architectures with a bottleneck layer, speeding up training.

### 5.2 NUMBER OF FUNCTION EVALUATIONS VS. DATA DIMENSION

The full computational cost of integrating the instantaneous change of variables (2) is $\mathcal{O}(DH\widehat{L})$ where $D$ is dimensionality of the data, $H$ is the size of the hidden state, and $\widehat{L}$ is the number of function evaluations (NFE) that the adaptive solver uses to integrate the ODE. In general, each evaluation of the model is $\mathcal{O}(DH)$ and in practice, $H$ is typically chosen to be close to $D$. Since the general form of the discrete change of variables equation (1) requires $\mathcal{O}(D^3)$-cost, one may wonder whether the number of evaluations $\widehat{L}$ depends on $D$.

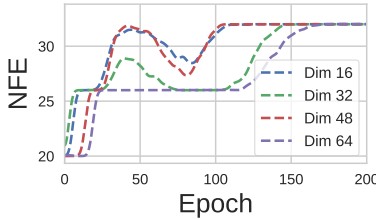

We train VAEs using FFJORD flows with increasing latent dimension $D$. The NFE throughout training is shown in Figure 5. In all models, we find that the NFE increases throughout training, but converges to the same value, independent of $D$. We conjecture that the number of evaluations is not dependent on the dimensionality of the data but the complexity of its distribution, or more specifically, how difficult it is to transform its density into the base distribution.

Figure 5: NFE used by the adaptive ODE solver is approximately independent of data-dimension. Lines are smoothed using a Gaussian filter.

### 5.3 SINGLE-SCALE VS. MULTI-SCALE FFJORD

Crucial to the scalability of Real NVP and Glow is the multiscale architecture originally proposed in Dinh et al. (2017). We compare a single-scale encoder-decoder style FFJORD with a multiscale FFJORD on the MNIST dataset where both models have a comparable number of parameters and plot the total NFE–in both forward and backward passes–against the loss achieved in Figure 6. We find that while the single-scale model uses approximately one half as many function evaluations as the multiscale model, it is not able to achieve the same performance as the multiscale model.

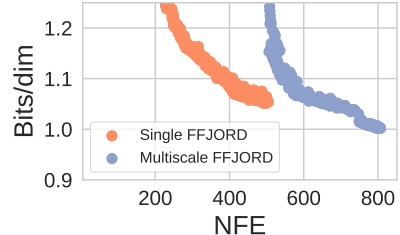

Figure 6: For image data, a single FFJORD flow can achieve near performance to multi-scale architecture while using half the number of evaluations.

## 6 SCOPE AND LIMITATIONS

**Number of function evaluations can be prohibitive.** The number of function evaluations required to integrate the dynamics is not fixed ahead of time, and is a function of the data, model architecture, and model parameters. This number tends to grow as the models trains and can become prohibitively large, even when memory stays constant due to the adjoint method. Various forms of regularization such as weight decay and spectral normalization (Miyato et al., 2018) can be used to reduce the this quantity, but their use tends to hurt performance slightly.

**Limitations of general-purpose ODE solvers.** In theory, our model can approximate any differential equation (given mild assumptions based on existence and uniqueness of the solution), but in practice our reliance on general-purpose ODE solvers restricts us to non-stiff differential equations that can be efficiently solved. ODE solvers for stiff dynamics exist, but they evaluate $f$ many more times to achieve the same error. We find that a small amount of weight decay regularizes the ODE to be sufficiently non-stiff.

## 7 CONCLUSION

We have presented FFJORD, a reversible generative model for high-dimensional data which can compute exact log-likelihoods and can be sampled from efficiently. Our model uses continuous-time dynamics to produce a generative model which is parameterized by an unrestricted neural network. All required quantities for training and sampling can be computed using automatic differentiation, Hutchinson's trace estimator, and black-box ODE solvers. Our model stands in contrast to other methods with similar properties which rely on restricted, hand-engineered neural network architectures. We demonstrated that this additional flexibility allows our approach to achieve on-par or improved performance on density estimation and variational inference.

We believe there is much room for further work exploring and improving this method. FFJORD is empirically slower to evaluate than other reversible models like Real NVP or Glow, so we are interested specifically in ways to reduce the number of function evaluations used by the ODE-solver without hurting predictive performance. Advancements like these will be crucial in scaling this method to even higher-dimensional datasets.

## 8 ACKNOWLEDGEMENTS

We thank Yulia Rubanova and Roger Grosse for helpful discussions.

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

## APPENDIX A   QUALITATIVE SAMPLES

Samples from our FFJORD models trained on MNIST and CIFAR10 can be found in Figure 7.

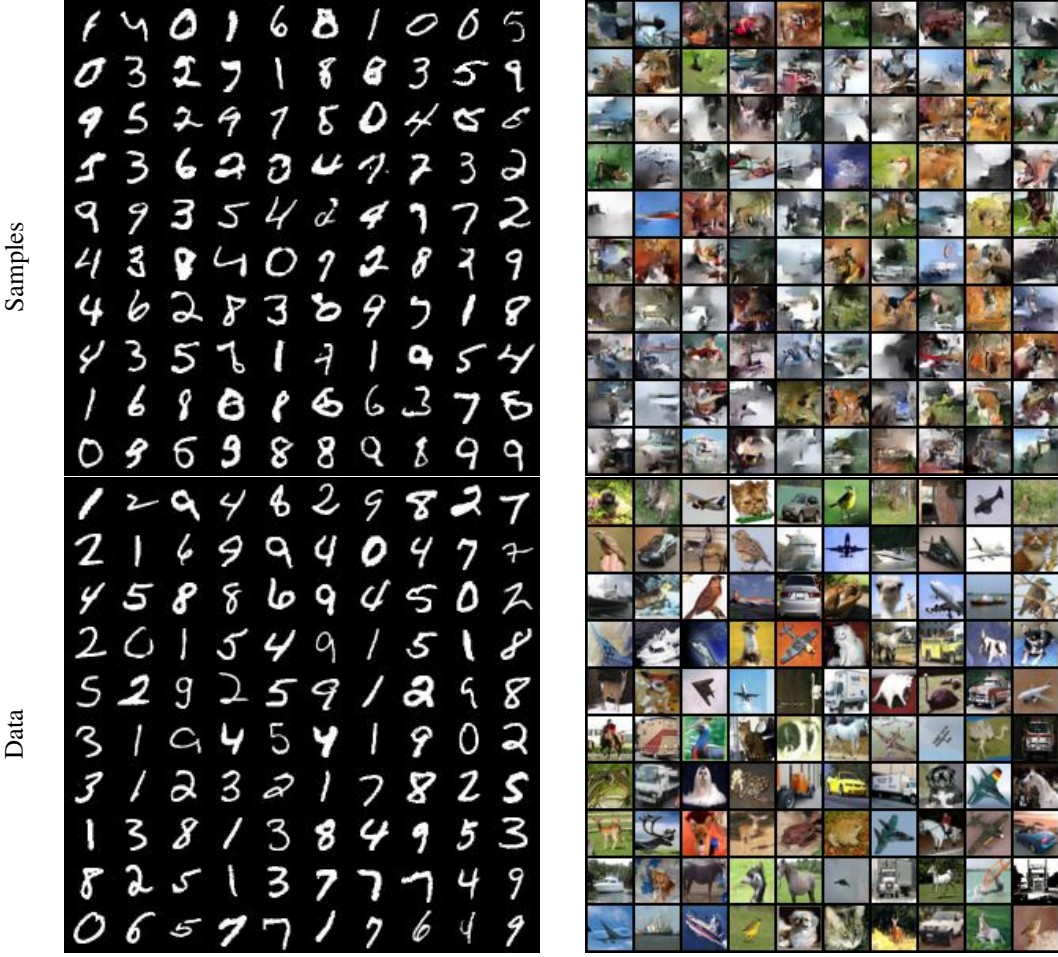

Figure 7: Samples and data from our image models. MNIST on left, CIFAR10 on right.

## APPENDIX B   EXPERIMENTAL DETAILS AND ADDITIONAL RESULTS

### B.1   DENSITY ESTIMATION

On the tabular datasets we performed a grid-search over network architectures. We searched over models with 1, 2, 5, or 10 flows with 1, 2, 3, or 4 hidden layers per flow. Since each dataset has a different number of dimensions, we searched over hidden dimensions equal to 5, 10, or 20 times the data dimension (hidden dimension multiplier in Table 4). We tried both the tanh and softplus nonlinearities. The best performing models can be found in the Table 4.

On the image datasets we experimented with two different model architectures; a single flow with an encoder-decoder style architecture and a multiscale architecture composed of multiple flows.

While they were able to fit MNIST and obtain competitive performance, the encoder-decoder architectures were unable to fit more complicated image datasets such as CIFAR10 and Street View House Numbers. The architecture for MNIST which obtained the results in Table 2 was composed of four convolutional layers with $64 \rightarrow 64 \rightarrow 128 \rightarrow 128$ filters and down-sampling with strided convolutions by two every other layer. There are then four transpose-convolutional layers who's

filters mirror the first four layers and up-sample by two every other layer. The softplus activation function is used in every layer.

The multiscale architectures were inspired by those presented in Dinh et al. (2017). We compose multiple flows together interspersed with "squeeze" operations which down-sample the spatial resolution of the images and increase the number of channels. These operations are stacked into a "scale block" which contains $N$ flows, a squeeze, then $N$ flows. For MNIST we use 3 scale blocks and for CIFAR10 we use 4 scale blocks and let $N = 2$ for both datasets. Each flow is defined by 3 convolutional layers with 64 filters and a kernel size of 3. The softplus nonlinearity is used in all layers.

Both models were trained with the Adam optimizer (Kingma & Ba, 2015). We trained for 500 epochs with a learning rate of .001 which was decayed to .0001 after 250 epochs. Training took place on six GPUs and completed after approximately five days.

## B.2 VARIATIONAL AUTOENCODER

Our experimental procedure exactly mirrors that of Berg et al. (2018). We use the same 7-layer encoder and decoder, learning rate (.001), optimizer (Adam Kingma & Ba (2015)), batch size (100), and early stopping procedure (stop after 100 epochs of no validaiton improvment). The only difference was in the nomralizing flow used in the approximate posterior.

We performed a grid-search over neural network architectures for the dynamics of FFJORD. We searched over networks with 1 and 2 hidden layers and hidden dimension 512, 1024, and 2048. We used flows with 1, 2, or 5 steps and wight matrix updates of rank 1, 20, and 64. We use the softplus activation function for all datasets except for Caltech Silhouettes where we used tanh. The best performing models can be found in the Table 5. Models were trained on a single GPU and training took between four hours and three days depending on the dataset.

| Dataset | nonlinearity | # layers | hidden dim multiplier | # flow steps | batchsize |
|---------|--------------|----------|-----------------------|--------------|-----------|
| POWER | tanh | 3 | 10 | 5 | 10000 |
| GAS | tanh | 3 | 20 | 5 | 1000 |
| HEPMASS | softplus | 2 | 10 | 10 | 10000 |
| MINIBOONE | softplus | 2 | 20 | 1 | 1000 |
| BSDS300 | softplus | 3 | 20 | 2 | 10000 |

Table 4: Best performing model architectures for density estimation on tabular data with FFJORD.

| Dataset | nonlinearity | # layers | hidden dimension | # flow steps | rank |
|---------|--------------|----------|------------------|--------------|------|
| MNIST | softplus | 2 | 1024 | 2 | 64 |
| Omniglot | softplus | 2 | 512 | 5 | 20 |
| Frey Faces | softplus | 2 | 512 | 2 | 20 |
| Caltech | tanh | 1 | 2048 | 1 | 20 |

Table 5: Best performing model architectures for VAEs with FFJORD.

B.3    STANDARD DEVIATIONS FOR TABULAR DENSITY ESTIMATION

|          | POWER | GAS | HEPMASS | MINIBOONE | BSDS300 |
|----------|-------|-----|---------|-----------|---------|
| Real NVP | -0.17 ± 0.01 | -8.33 ± 0.14 | 18.71 ± 0.02 | 13.55 ± 0.49 | -153.28 ± 1.78 |
| Glow | -0.17 ± 0.01 | -8.15 ± 0.40 | 18.92 ± 0.08 | 11.35 ± 0.07 | -155.07 ± 0.03 |
| FFJORD | -0.46 ± 0.01 | -8.59 ± 0.12 | 14.92 ± 0.08 | 10.43 ± 0.04 | -157.40 ± 0.19 |
| MADE | 3.08 ± 0.03 | -3.56 ± 0.04 | 20.98 ± 0.02 | 15.59 ± 0.50 | -148.85 ± 0.28 |
| MAF | -0.24 ± 0.01 | -10.08 ± 0.02 | 17.70 ± 0.02 | 11.75 ± 0.44 | -155.69 ± 0.28 |
| TAN | -0.48 ± 0.01 | -11.19 ± 0.02 | 15.12 ± 0.02 | 11.01 ± 0.48 | -157.03 ± 0.07 |
| MAF-DDSF | -0.62 ± 0.01 | -11.96 ± 0.33 | 15.09 ± 0.40 | 8.86 ± 0.15 | -157.73 ± 0.04 |

Table 6: Negative log-likehood on test data for density estimation models. Means/stdev over 3 runs. Real NVP, MADE, MAF, TAN, and MAF-DDSF results on are taken from Huang et al. (2018). In reproducing Glow, we were able to get comparable results to the reported Real NVP by removing the invertible fully connected layers.

APPENDIX C    NUMERICAL ERROR FROM THE ODE SOLVER

ODE solvers are numerical integration methods so there is error inherent in their outputs. Adaptive solvers (like those used in all of our experiments) attempt to predict the errors that they accrue and modify their step-size to reduce their error below a user set tolerance. It is important to be aware of this error when we use these solvers for density estimation as the solver outputs the density that we report and compare with other methods. When tolerance is too low, we run into machine precision errors. Similarly when tolerance is too high, errors are large, our training objective becomes biased and we can run into divergent training dynamics.

Since a valid probability density function integrates to one, we take a model trained on Figure 1 and numerically find the area under the curve using Riemann sum and a very fine grid. We do this for a range of tolerance values and show the resulting error in Figure 8. We set both `atol` and `rtol` to the same tolerance.

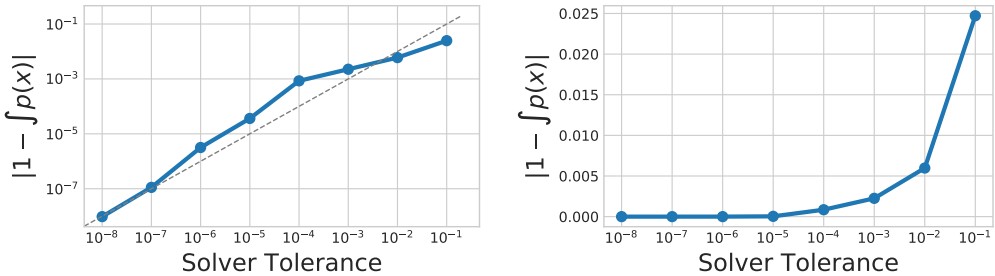

Figure 8: Numerical integration shows that the density under the model does integrate to one given sufficiently low tolerance. Both log and non-log plots are shown.

The numerical error follows the same order as the tolerance, as expected. During training, we find that the error becomes non-negligible when using tolerance values higher than $10^{-5}$. For most of our experiments, we set tolerance to $10^{-5}$ as that gives reasonable performance while requiring few number of evaluations. For the tabular experiments, we use `atol`=$10^{-8}$ and `rtol`=$10^{-6}$.

