# OpenReview forum: "FFJORD: Free-Form Continuous Dynamics for Scalable Reversible Generative Models"
_ICLR.cc/2019/Conference_

### Official Review · AnonReviewer1 · 2018-11-01
**Scaling up of Neural ODE model for generative model**

**Rating:** 7
**Confidence:** 3

**Review:**

This paper further explores the work of Chen et al. (2018) applied to reversible generative modelling. While section 1 and 2 focuses on framing the context of this work. The ODE solver architecture for continuous normalizing flow learn a density mapping using an instantaneous change of variable formula.
The contribution of this work seems to be enabling the use of deeper neural network than in Chen et al. (2018)  as part of the ODE solver flow. While the single-layer architecture in Chen et al. (2018) enable efficient exact computation of the Jacobian Trace, using a deeper architecture compromises that property. As a result, the authors propose to use the unbiased Hutchinson trace estimator of the Jacobian Trace. Furthermore, the authors observe that using a bottleneck architecture reduces the rank of the Jacobian and can therefore help reducing the variance of the estimator.
The density estimation task in 2D is nice to see but lacks comparison with Chen et al. (2018), on which this paper improves. Moreover, is the Glow model used here only using additive coupling layers? If so, this might explain the difficulties of this Glow model.
Although the model presented in this paper doesn't obtain state-of-the-art results on the larger problems, the work presented in this paper demonstrates the ability of ODE solvers as continuous normalizing flows to be competitive in the space of prescribed model.
Concerning discussions and analysis:
- given the lack of improvement using the bottleneck trick, is there an actual improvement in variance using this trick? or is this trick merely explaining why using a bottleneck architecture more suited for the Hutchinson trace estimator?
In algorithm 1, is \epsilon only one random vector that keeps being reused at every step of the solver algorithm? I would be surprised that the use of a single random vector across different steps did not significantly increased the variance of the estimator.

---

> ### Author Response · Authors · 2018-11-15
> **Response to reviewer 1**
>
> Thank you very much for your kind words about our work. We will address you comments in order.
>
> We have added a comparison to Chen et al. (2018) in our 2D density estimation experiments. Our model performs favorably compared to this baseline.
>
> We note that the Glow model we used in all of our density estimation experiments used affine transformation layers with both learned scale and translation. We have added a note to the appendix to clarify this.
>
> Regarding the state-of-the-art performance of the model we would like to stress that or model is mainly comparable to Glow and Real NVP since it is a reversible flow-based generative model with efficient sampling. When compared to models in this class, FFJORD performs the best by a wide margin on all datasets tested except for CIFAR10. The models in the lower half of table 2 are not directly comparable to FFJORD since they are autoregressive and cannot be efficiently sampled from (some cannot even be analytically sampled from). We include those models to demonstrate that FFJORD performs comparably to these models (and outperforms them on some datasets) for density estimation but also has the additional ability to be sampled from and inverted.
>
> Regarding the bottleneck variance reduction we note that we do observe reduced variance as can be seen in figure 4. However, we did find that the variance of Hutchinson’s estimator did not provide any problems when training FFJORD models so we simply used it as is in our experiments. We also did notice that using bottleneck layers tended to reduce performance. We included a discussion of the bottleneck trick to provide the community with some ideas on how to deal with the variance that the estimator adds if it becomes a problem for any future practitioners implementing our method. We have added a sentence to section 3.1.1 to make this more clear.
>
> To clarify, we use a single epsilon for each integration. This can be seen in the last line of equation 8 where the integral is inside of the expectation indicating that we first sample epsilon, then integrate (epsilon^T * (df/dz) * epsilon). Resampling epsilon during every step of the numerical solver would mean we’re solving a random ordinary differential equation, which our solvers are not equipped to handle; it would dramatically increase numerical instability.
>
> We thank you for your review and hope you will appreciate the changes that it has inspired in our paper.

---

### Official Review · AnonReviewer3 · 2018-11-02
**generative modeling with ODE's and Hutchinson's trace estimator, decent paper.**

**Rating:** 7
**Confidence:** 4

**Review:**

Summary:
This paper discusses an advance in the framework of normalizing flows for generative modeling, named FFJORD. The authors consider normalizing flows in the form of ordinary differential equations, as also discussed in [1]. Their contributions are two-fold: (1) they use an unbiased estimator of the likelihood of the model by approximating the trace of the jacobian with Hutchinson’s trace estimator, (2) they have implemented the required ODE solvers on GPUs.

The models are evaluated on a density estimation task on tabular data and two image datasets (MNIST and CIFAR10), as well as on variational inference for auto-encoders, where the datasets MNIST, Omniglot, Freyfaces and Caltech Silhouettes are considered.

The authors argue that the trace estimator, in combination with reverse-mode automatic differentiation to compute vector-Jacobian products, leads to a computational cost of O(D), instead of O(D^2) for the exact trace of the jacobian.
They compare this to the cost of computing a Jacobian determinant for finite flows, which is O(D^3) in general. They argue that in general all works on finite flows have adjusted their architectures for the flows to avoid the O(D^3) complexity, and that FFJORD has no such restriction.
However, I would like the authors to comment on the following train of thought: autoregressive models, such as MAF, as well as IAF (inverse of an autoregressive model) do not require O(D^3) to compute jacobian determinants as the jacobian is of triangular form. Note however, they are still universal approximators if sufficient flows are applied, as any distribution can be factorized in an autoregressive manner. With this in mind, I find the red cross for MAF under free-form Jacobian slightly misleading. Perhaps I misunderstood something, so please clarify.

Another topic that I would like the authors to comment on is efficiency and practical use. One of the main points that the authors seem to emphasise, is that contrary to autoregressive models, which require D passes through the model to sample a datapoint of size D, FFJORD is a ‘single-pass’ model, requiring only one pass through the model. They therefore indicate that they can do efficient sampling. However, for FFJORD every forward pass requires a pass through an ODE solver, which as the authors also state, can be very slow. I could imagine that this is still faster than an autoregressive model, but I doubt this is actually of comparable speed to a forward pass of a finite flow such as glow or realNVP.
On the other hand, autoregressive models do not require D passes during training, whereas, if I understand correctly, FFJORD relies on two passes through ODE solvers, one for computing the loss, and a second to compute the gradient of the loss with respect to model parameters. So autoregressive models should train considerably faster. The authors do comment on the fact that FFJORD is slower than other models, but they do not give a hint as to how much slower it is. This would be of importance for practical use, and for other people to consider using FFJORD in future work.

For the density estimation task, FFJORD does not have the best performance compared other baselines, except for MNIST, for which the overall best model was not evaluated (MAF-DDSF). For variational inference FFJORD is stated to outperform all other flows, but the models are only evaluated on the negative evidence lower bound, and not on the negative log-likehood (NLL). I suspect the NLL to be absent from the paper as it requires more computation, and this takes a long time for FFJORD. Without an evaluation on NLL the improvement over other methods is questionable. Even if the improvement still holds for the NLL, the relative improvement might not weigh heavily enough against increased runtime. FFJORD does require less memory than its competitors.

The improved runtime by implementing the ODE solvers on GPU versus the runtime on a CPU would be useful, given that this is listed as one of the main contributions.

Besides these questions/comments, I do think the idea of using Hutchinsons trace estimator is a valid contribution, and the experimental validation of continuous normalizing flows is of interest to the research community. Therefore, in my opinion, the community will benefit from the information in this paper, and it should be accepted. However I do wish for the authors to address the above questions as it would give a clearer view of the practical use of the proposed model.

See below for comments and questions:

Quality
The paper has a good setup, and is well structured. The scope and limitations section is very much appreciated.

Clarity
The paper is clearly written overall. The only section I can comment on is the related work section, which is not the best part of the paper. The division in normalizing flows and partitioned transformations is a bit odd. Partitioned transformations surely are also normalizing flows. Furthermore IAF by Kingma et al. is put in the box of autoregressive models, whereas it is the inverse of an autoregressive model, such that it does not have the D-pass sample problem. For a reader who is not too familiar with normalizing flows literature, I think this section is a little confusing. Furthermore, there is no related work discussed on continuous time flows, such as (but not limited to) [2].

Originality
The originality of the paper is not stellar, but sufficient for acceptance.

Significance
The community can benefit from the experimental analysis of continuous time flows, and the GPU implementation of the ODE solver. Therefore I think this work is significant.

Detailed questions/comments:

1. In section 4.2, as an additional downside to MAF-DDSF, the authors argue that sampling cannot be performed analytically. Since FFJORD needs to numerically propagate the ODE, I do not think FFJORD can sample analytically either. Is this correct?
2. The authors argue that they have no restriction on the architecture of the function f, even if they have O(D) estimation of the trace of the jacobian. However, they also say they make use of the bottle-neck trick to reduce the variance that arises due to Hutchinson’s estimate of the trace. This seems like a limitation on the architecture to me. Can the authors comment?
3. In B.1 in the appendix, the street view house numbers dataset is mentioned, but no results appear in the main text, why not?
4. In the results section, it is not clear to me which numbers of the baselines for different datasets are taken from other papers, and which numbers are obtained by the authors of this paper. Please clarify.
5. In the conclusions, when discussing future work, the authors state that they are interested in reducing the number of function evaluations in the ODE solvers. In various disciplines many people have worked on this problem for a long time. Do the authors think major improvements are soon to be made?
6. In section 5.2 the dependence of the number of function evaluations (NFE) on the data dimension D is discussed. As a thought experiment they use the fact that going from an isotropic gaussian distribution (in any D), to an isotropic gaussian distribution has a corresponding differential equation of zero. This should convince the reader that NFE is independent of D. However, this seems to me to be such a singular example, that I gain no insight from it, and it is not very convincing. Do the authors agree that this particular example does not add much? If not, please explain.

[1] Chen et al. Neural ordinary differential equations. NIPS 2018
[2] Chen et al. Continuous-time flows for deep generative models.

**** EDIT *****

I have read the response of the authors and appreciate their clarifications and the additional information on the runtimes. See my response below for the concern that remains about the absence of the estimate of the log likelihood for the VAE experiments. Besides this issue, the other comments/answers were satisfactory, and I think this paper is of interest to the research community, so I will stick with my score.

---

> ### Author Response · Authors · 2018-11-15
> **Response to reviewer 3**
>
> Response:
>
> We thank the reviewer for their thoughtful comments and questions. We address them in order.
>
> You ask us to clarify why we have said that autoregressive models do not have a free-form jacobian. As you mention, each step of flow in these models is restricted to have a triangular jacobian. While these models can be universal density estimators, their underlying neural network components must be restricted to allow for efficient training. For this reason, we say they do not have a free-form jacobian.
>
> FFJORD requires the use of a numerical ODE solver and it is true that for each step of the solver, a forward pass must be computed through the gradient function f(z, t) = dz/dt but we demonstrate in figure 5 that the number of these forward evaluations is not a function of D in practice so it can be considered a “one-pass” model.
>
> Regarding efficiency of training, FFJORD requires two calls to an ODE solver. One to compute the z_t and one to compute the gradients. This is analogous to the standard backpropagation algorithm which requires one forward pass to compute the intermediate activations and one backward pass to compute the gradients.
>
> We have added a more clarification regarding the speed of FFJORD compared to competing approaches in the conclusion.
>
> Regarding comparisons to other methods on density estimation we split table 2 into two sections. The top section consists of reversible generative  models (like FFJORD) and the bottom section contains autoregressive models. We believe FFJORD is more fairly comparable to reversible models like Real NVP and Glow (which FFJORD outperforms on all datasets except CIFAR10). We add the results in the bottom half to demonstrate that FFJORD performs comparably to powerful autoregressive models (unlike other reversible generative models) while also being efficient to sample from.
>
> Regarding the background section on partitioned transformation and normalizing flows, we decided to separate out partitioned transformations since they have been successfully used on their own to build large-scale generative models where other types of normalizing flow have only been used successfully in conjunction with variational autoencoders.
>
> 1) We will clarify this in the text but we would argue that FFJORD does have a known analytical inverse that is written as an integral. It is true we don’t solve this integral analytically--which we will clarify in the text--but in the case of MAF-DDSF the inverse is simply not known.
>
> 2) We place no restrictions on the architecture that we use in our experiments. This can be seen in Appendix B. We do not utilize the bottleneck trick in our experiments. We found the additional variance added by Hutchinson’s estimator does not negatively impact training at all. We present the bottleneck trick to give the reader an approach they can use to reduce the variance if this becomes problematic for them if they implement our method. We have added a sentence to section 3.1.1 to clarify this.
>
> 3) We initially used SVHN as a middle ground dataset which was more difficult than MNIST but less difficult than CIFAR10. We did not include quantitative results on this dataset since it is not a widely reported benchmark. We have removed all references to the dataset.
>
> 4) We have added a note in the appendix clarifying this, where we also report standard deviations for table 2.
>
> 5) This is a very widely studied problem but most of the existing research in this area is not easily applicable to our problem. The ODEs we are dealing with are of much higher dimension than those studied heavily in the literature on numerical ODE methods. Moreover, we have unique ODE structures which are not typically explored in numerical methods literature. An example of such a structure is that we are integrating a mini-batch of data through our ODE which has the dynamics defined by the same function f. It is possible that computation in this can be better reused. We hope that the numerical methods community will now become interested in these types of problems.
>
> 6) While yes, it is a simple example, we felt that it was sufficient since we also have experimental results that back up this claim.

---

### Official Review · AnonReviewer2 · 2018-11-02
**Nice algorithm but incremental**

**Rating:** 7
**Confidence:** 4

**Review:**

This paper discusses a technique for continuous normalization flow in which the transformations are not required to be volume preserving (the transformation with unity Jacobian), and architecture of neural network does not need to be designed to hold such property. Instead authors proposed no restriction on architecture of neural network to design their reversible mapping.
The Paper has good background and literature review, and as authors mentioned  this paper is base on the idea of  Chen, Tian Qi, et al. "Neural Ordinary Differential Equations." arXiv preprint arXiv:1806.07366 (2018). Chapter two of this paper is summary of  "Neural Ordinary Differential Equations." and chapter Three is main contribution of this paper that can be summarized under two points:

1- Authors borrowed the "continuous normalizing flow " in Chen et al. and they have designed unbiased log density estimator using Hutchinson trace estimator and evaluated the trace with complexity of O(D) (dimension of data) instead of O(D^2) that  is used in chen et al. Paper

2- They proposed by reducing the hidden layer dimension of neural network, it is possible that variance of estimator to be reduced

Novelty and Quality:
the main contribution of this paper is summarized above.
The paper do not contain any significant theorem or mathematical claims, it is more focused on design of linear algorithm that estimate continuous normalizing flow that they have borrowed from the Chen et al. paper.  This is a good achievement that can help continuous normalizing flow scale on data with higher dimensions, but in results and experiments section no comparison has been made to performance of chen et al. Also no guarantees or bound has been given about the variance reduction of  estimator and it is more based on the authors intuition.

Clarity:
The paper is well written and previous relevant methods have been reviewed well. There are a few issues that are listed below:
1-in section 3 the reason that dimensionality of estimator can reduce to D from D^2 can be explained more clearly

2- Figure 1 is located on first page of the paper but it has never been referred in main paper, just it is mentioned once in appendix , it can be moved to appendix.

3- in section 3.1.1 the “view view” can be changed to “view”

significance and experiments:
The experiments are very detailed and extensive and authors have compared their algorithm with many other competing algorithms and showed improvement in many of the cases.
As mentioned in Quality and Novelty part of the review, just one comparison is missing and that is the comparison to method that the paper is inspired by. It would be interesting to see how much trace estimator approach that has been used in this paper, would sacrifice the negative log-likelihood or ELBO specially in real data like MNIST and CIFAR 10.  it seems original paper has not reported the performance on those data-sets as well, is this difficult as chen et. al. paper algorithm for trace calculation has complexity of O(D^2)?

---

> ### Author Response · Authors · 2018-11-15
> **Response to reviewer 2**
>
> We thank the reviewer for their time and their kind word about our work. We will address your concerns and questions in the order you wrote them.
>
> You mention that we do not compare to Chen et al. We chose not to compare directly with this method because CNFs, as presented in their paper, should not be expected to scale to high dimensional data. The analogous comparison in discrete-time flows would be comparing Glow to stacked planar flows. We have added a comparison to Chen et al. On the 2D datasets which illustrates this point.
>
> The variance of Hutchinson’s estimator is well understood and we do note the asymptotic variance of the estimator in section 3.1.1. While we do not prove this trick reduces variance, we do demonstrate this empirically which can be seen in section 5.1.
>
> Regarding the “dimensionality” going from D^2 -> D. We believe you mean computation, correct? We believe this is clearly explained in section 3.1 which introduces the estimator that allows the computation to be reduced.
>
> We have added a note in the introduction to explain Figure 1.
> We have fixed the typo noticed in section 3.1.
>
> Regarding a comparison to Chen et al., we chose not to include one because while Chen et al. proposed CNFs and the objective that they optimize, they did not really present a generative model that uses this framework. The CNF they presented is comparable to the planar flows first presented in Rezende et al. (2015). These are fairly weak transformation and will not easily scale to the high-dimensional datasets we experimented with. An analogy would be comparing Glow to a stack of planar flows.
>
> You are correct in your reasoning about why we did not present log-likelihoods for the VAE experiments. While Hutchinson’s trace estimator gives us unbiased estimates of the ELBO, using it to estimate the log-likelihood with importance sampling gives an upper-bound due to the stochasticity of the estimator. It would be possible to use the brute-force Jacobian to estimate this, but the computation of doing so proved to be prohibitive.

---

> > ### Comment · AnonReviewer2 · 2018-11-29
> > **Thanks for response**
> >
> > I got answers to all questions I had by reviewing comments to my concerns and other reviewers'.  I have no further question and keep my score same. Thanks

---

### Public Comment · ~Jonas_Degrave1 · 2018-10-05
**Convergence of Hutchinson’s estimator analysis**

I find this paper and its predecessor a game changer and I am happy to see this more detailed analysis on backpropagating ODE's for density estimation. https://arxiv.org/pdf/1806.07366.pdf
I wrote an implementation of the neural ODE with its gradient in tensorflow, and was saddened to see that while I initially estimated it's speed at O(D) (from the naive O(D^3)), it was actually O(D^2) as the trace is indeed expensive. This unfortunately makes it considerably slower than for example a Real NVP, which is O(D). I did notice Hutchinson’s trace estimator might improve this neural ODE to O(D) again, but have not gone around to verify that unlike in this paper.

However, the bottleneck trick seems to undermine the original strength of the approach (allowing for wide networks). I wonder if you have an analysis to show the relation between the number of samples in equation 8 and the error in the likelihood. Specifically I wonder if it would be possible to train using only a single sample for the expectation in equation 8? Do you have an analysis on the amount of samples and how the estimate converges? While introducing considerable noise, it would speed up the training phase allowing big batches again. In the testing phase, evaluating a big batch is typically less of a problem and there more samples could be used anyway.

Otherwise, I admire this approach and hope to see more work in this direction in the future.

---

> ### Author Response · Authors · 2018-11-15
> **Thanks for the comments**
>
> Thank you for your kind words about our paper. Your experience is very similar to ours in developing this method. Regarding the bottleneck, yes, we agree that using using this type of architecture will in general produce a weaker model. An updated version of our paper clarifies this. In our experiments we did not actually use this because we found that using wide networks gives better performance and a single sample to estimate the expectation from equation 8 worked fine in training our models.

---

### Public Comment · ~Manzil_Zaheer1 · 2018-12-08
**Discrepancy in Likelihood Values**

It looks like the authors are not reporting the most up-to-date likelihoods using TANs (as per the Table 1 in official ICML paper http://proceedings.mlr.press/v80/oliva18a.html ). Hence the numbers reported in Table 2 in the paper should be updated.

---

> ### Author Response · Authors · 2018-12-14
> **Thanks**
>
> Thank for you pointing this out. We will update the camera-ready version with the correct results if our paper is accepted.

---

### Public Comment · ~Octavian_Eugen_Ganea1 · 2019-02-21
**free form Jacobian vs invertible functions**

Nice work! I would appreciate if you could shed some light into the following: you say that the Jacobian of f has free form, but how does this ensure invertibility ? You cite a result from Khalil,2002 stating that is is enough for f to be uniformly Lipschitz for the above to hold - can you please refer to the exact book chapter where I can find this ? Thanks.

---

### Public Comment · ~Thanh_Tung_Hoang1 · 2019-04-01
**How does FFJORD deal with overfitting?**

Dear authors,
Thank you for your nice work. I would appreciate if you could share your opinions on the generalization of FFJORD.

As I understand from your paper, FFJORD maximizes the log density of the data as shown in Equation 3. One way to maximize it is to select z's with higher density p(z). For diagonal Gaussians, these are points located around the mean. So is that possible for two very similar inputs z_1 and z_2 to be mapped to two very different outputs x_1 and x_2. The problem can be considered as the inverse of the mode collapse problem in GANs. That mental experiment raises another question: What are the properties of the landscape of generated distribution? Will it contain many sharp peaks and valleys and the model will just remember the training dataset? Do you have any measures to prevent FFJORD from overfitting to the training dataset? Thanks.

---

### Meta-Review · Area_Chair1 · 2018-12-15

**Confidence:** 4
**Recommendation:** Accept (Oral)

**Metareview:**

This paper proposes the use of recently propose neural ODEs in a flow-based generative model.

As the paper shows, a big advantage of a neural ODE in a generative flow is that an unbiased estimator of the log-determinant of the mapping is straightforward to construct. Another advantage, compared to earlier published flows, is that all variables can be updated in parallel, as the method does not require "chopping up" the variables into blocks.  The paper shows significant improvements on several benchmarks, and seems to be a promising venue for further research.

A disadvantage of the method is that the authors were unable to show that the method could produce results that were similar (of better than) the SOTA on the more challenging benchmark of CIFAR-10. Another downside is its computational cost. Since neural ODEs are relatively new, however, these problems might resolved with further refinements to the method.